# Comparative Transcriptomic Analysis of Insecticide-Resistant *Aedes aegypti* from Puerto Rico Reveals Insecticide-Specific Patterns of Gene Expression

**DOI:** 10.3390/genes14081626

**Published:** 2023-08-15

**Authors:** Dieunel Derilus, Lucy Mackenzie Impoinvil, Ephantus J. Muturi, Janet McAllister, Joan Kenney, Steven E. Massey, Ryan Hemme, Linda Kothera, Audrey Lenhart

**Affiliations:** 1Entomology Branch, Division of Parasitic Diseases and Malaria, Centers for Disease Control and Prevention, Atlanta, GA 30329, USA; ykd8@cdc.gov (L.M.I.); ephantus.muturi@usda.gov (E.J.M.); 2Division of Vector-Borne Diseases, Centers for Disease Control and Prevention, Fort Collins, CO 80521, USA; jvm6@cdc.gov (J.M.); vwx1@cdc.gov (J.K.); fph6@cdc.gov (L.K.); 3Biology Department, University of Puerto Rico-Rio Piedras, San Juan, PR 00925, USA; stevenemassey@gmail.com; 4Division of Vector-Borne Diseases, Centers for Disease Control and Prevention, San Juan, PR 00920, USA; wma0@cdc.gov

**Keywords:** *Aedes aegypti*, insecticide resistance, RNA-Seq, detoxification genes, mosquitoes, Puerto Rico, organophosphate resistance, pyrethroid resistance

## Abstract

*Aedes aegypti* transmits major arboviruses of public health importance, including dengue, chikungunya, Zika, and yellow fever. The use of insecticides represents the cornerstone of vector control; however, insecticide resistance in *Ae. aegypti* has become widespread. Understanding the molecular basis of insecticide resistance in this species is crucial to design effective resistance management strategies. Here, we applied Illumina RNA-Seq to study the gene expression patterns associated with resistance to three widely used insecticides (malathion, alphacypermethrin, and lambda-cyhalothrin) in *Ae. aegypti* populations from two sites (Manatí and Isabela) in Puerto Rico (PR). Cytochrome P450s were the most overexpressed detoxification genes across all resistant phenotypes. Some detoxification genes (CYP6Z7, CYP28A5, CYP9J2, CYP6Z6, CYP6BB2, CYP6M9, and two CYP9F2 orthologs) were commonly overexpressed in mosquitoes that survived exposure to all three insecticides (independent of geographical origin) while others including CYP6BY1 (malathion), GSTD1 (alpha-cypermethrin), CYP4H29 and GSTE6 (lambda-cyhalothrin) were uniquely overexpressed in mosquitoes that survived exposure to specific insecticides. The gene ontology (GO) terms associated with monooxygenase, iron binding, and passive transmembrane transporter activities were significantly enriched in four out of six resistant vs. susceptible comparisons while serine protease activity was elevated in all insecticide-resistant groups relative to the susceptible strain. Interestingly, cuticular-related protein genes (chinase and chitin) were predominantly downregulated, which was also confirmed in the functional enrichment analysis. This RNA-Seq analysis presents a detailed picture of the candidate detoxification genes and other pathways that are potentially associated with pyrethroid and organophosphate resistance in *Ae. aegypti* populations from PR. These results could inform development of novel molecular tools for detection of resistance-associated gene expression in this important arbovirus vector and guide the design and implementation of resistance management strategies.

## 1. Introduction

Chemical insecticides are widely used to control insects of public health significance. However, the intensive and widespread use of these insecticides has led to rapid and widespread evolution of insecticide resistance in major insect disease vectors, raising concerns about their sustained use in public health. Developing strategies to maintain efficacy of the limited number of chemical insecticides approved for public health use in the face of growing insecticide resistance is a major priority for the control of mosquito-borne diseases. This effort could benefit from an in-depth understanding of the molecular basis of resistance to key insecticides used in public health vector control.

In 2016, a Zika outbreak was declared in Puerto Rico [1]. In response to this Zika outbreak, a rapid screening of *Ae. aegypti* for susceptibility to key insecticides used for mosquito control was conducted in 38 locations across Puerto Rico, and results showed widespread resistance to pyrethroids and malathion [2]. However, little is known about the molecular basis of insecticide resistance in *Ae. aegypti* in Puerto Rico.

Target site insensitivity and increased metabolic detoxification are the main mechanisms responsible for mosquito tolerance to insecticides, but other mechanisms such as behavioral avoidance of insecticide-treated surfaces and cuticular alterations have also been reported [3,4,5,6]. Target site insensitivity occurs when a mutation of the protein targeted by the insecticide prevents the insecticide from binding to the molecule. These mutations are well characterized in mosquitoes, and molecular tests for their detection are available for many major vectors [7,8,9]. In contrast, metabolic resistance is much less understood and is primarily conferred via gene amplifications, transcriptional enhancements, and coding mutations in detoxification enzymes, particularly cytochrome P450 monooxygenases, carboxylesterases, and glutathione-S-transferases [10]. As metabolic resistance can arise from multiple mechanisms and involves many gene families, understanding its molecular basis remains a complex public health challenge. Moreover, the specific detoxification genes involved in metabolic resistance can vary greatly between insecticides and across mosquito populations and species [11,12,13]. These complexities hamper the development of universal molecular diagnostic markers for metabolic resistance and highlight the need for site-specific understanding of the molecular mechanisms responsible for insecticide resistance to facilitate development of evidence-based resistance management strategies. 

Here, we use RNA-Seq to examine the gene expression profiles of malathion, alpha-cypermethrin, and lambda-cyhalothrin resistant populations of *Ae. aegypti* from two locations in Puerto Rico. This mosquito species lives near humans in urban settings and is the primary vector of several globally important arthropod-borne viruses (arboviruses) including dengue, chikungunya, Zika, and yellow fever. Resistance of *Ae. aegypti* to multiple insecticide classes has been reported worldwide, posing serious operational challenges towards its control [14,15,16,17]. Both target site insensitivity and metabolic resistance have been identified as important mechanisms of insecticide resistance in *Ae. aegypti,* and several previous studies have investigated the specific genes involved in metabolic detoxification of insecticides in this mosquito species [18,19,20,21,22,23,24,25,26,27,28]. This study aimed to disentangle the transcriptional basis of resistance to multiple insecticides in *Ae. aegypti* from two geographically distinct populations, with the aim of better understanding the specificity of metabolic resistance with respect to insecticide and geography.

## 2. Materials and Methods

### 2.1. Study Area and Mosquito Collections

Two populations of *Ae. aegypti* were used in this study, collected in the spring of 2017 from the towns of Isabela (18.4704° N, 67.0242° W) and Manatí (18.4330° N, 66.4759° W) in Puerto Rico. Isabela and Manatí are located approximately 43 miles apart, and mosquitoes collected from these two sites had previously exhibited different levels of resistance to both malathion and pyrethroids [2]. Eggs were collected from the adults and shipped to the U.S. Centers for Disease Control and Prevention (CDC) in Atlanta, USA. At CDC, the mosquitoes were reared at 27 ± 2 °C, 70 ± 10% relative humidity, and a 14L:10D photoperiod. Adults were provided 10% sucrose ad libitum. The Rockefeller (ROCK) reference susceptible strain (obtained from the Malaria Research and Reference Reagent Resource Center, MR4) was used as a susceptible control. All three strains were reared in parallel for each of the biological replicates for the RNA-Seq libraries.

### 2.2. Insecticide Bioassays

Three-to-four-day old non-blood fed and unmated *Ae. aegypti* females from Isabela (ISA) and Manatí (MAN) were tested for resistance to diagnostic doses of alpha-cypermethrin (10 μg/bottle), malathion (50 μg/bottle), and lambda-cyhalothrin (10 μg/bottle) using the CDC bottle bioassay [29]. Assays were carried out in four replicates, each containing approximately 25 individuals per bottle. For each test, a control bottle coated with only 1 mL of acetone was included. Mosquitoes alive after 30 min of insecticide exposure were considered resistant. Assays were repeated to obtain sufficient numbers of phenotyped mosquitoes for RNA extraction. For the susceptible ROCK strain, three-to-four-day-old adult non-blood fed and unmated female mosquitoes were killed by freezing and stored at −80 °C until RNA extraction was conducted. For ISA and MAN, mosquitoes that were alive (survivors) at the end of the 30 min exposure period were separated and tallied, then frozen at −80 °C. In addition, individuals from ISA and MAN that were used in the control bottle and not exposed to insecticide were killed by freezing and stored at −80 °C for RNA extraction and sequencing.

### 2.3. RNA and Library Preparation Methods

Three biological replicates with pools of five mosquitoes each were prepared for three groups of mosquitoes: the susceptible *Ae. aegypti* Rockefeller strain (ROCK), field-collected mosquitoes that were not exposed to insecticides during the bottle bioassay, and field-collected mosquitoes that survived exposure to insecticides at the diagnostic dose and diagnostic time during the bottle bioassay. RNA was extracted using the Applied Biosystems Arcturus PicoPure RNA Isolation Kit (Arcturus, Applied Biosystems, Carlsbad, CA, USA) and quantified using the Agilent RNA ScreenTape 4200 Assay, according to the manufacturer’s protocols. Two micrograms of starting material were treated with Baseline-ZERO™ DNase (Lucigen, Middleton, WI, USA), and ribosomal RNA was removed using the Ribo-Zero™ Magnetic Core Kit and Ribo-Zero™ rRNA Removal Kit (Illumina, San Diego, CA, USA), according to the manufacturer’s protocols. RNA-Seq libraries were prepared from each pool of extracted RNA using the ScriptSeq™ v2 RNA-Seq Library Preparation Kit (Illumina, IL, USA), using 12 cycles of PCR amplification, according to the manufacturer’s protocol. Libraries were purified using Agencourt AMPure XP beads (Beckman Coulter, Brea, CA, USA) and assessed for quantity and size distribution using the Agilent DNA ScreenTape D5000 Assay. Each library was barcoded, and equal amounts of each library were pooled and sequenced (2 × 125 bp paired end) on an Illumina HiSeq 2500 sequencer, using v2 chemistry. Sequencing was performed at the Biotechnology Core Facility at CDC, Atlanta, GA, USA.

### 2.4. Measurement of Gene Expression Using Quantitative PCR

Five genes that were significantly differentially expressed in the RNA-Seq analysis were validated using quantitative PCR (qPCR). One microgram of RNA from three replicates of samples resistant to lambda-cyhalothrin from Manatí and Isabela and samples resistant to alphacypermethrin from Isabela were used to synthesize cDNA using the High-Capacity cDNA Reverse Transcription Kit (Applied Biosystems) with oligo-dT20 (NEB), according to the manufacturer’s instructions. RNA and cDNA synthesis also occurred in the susceptible laboratory strain ROCK. There was insufficient material remaining from the unexposed samples from both locations; hence, this was not included in the qPCR validation. The primers used are listed in Appendix A. Standard curves of Ct values for each gene were generated using a serial dilution of cDNA, allowing assessment of PCR efficiency. The qPCR amplification was carried out on a QuantStudio 6 Flex Real-Time PCR system (Applied Biosystems) using PowerUp SYBR Green Master Mix (Applied Biosystems). cDNA from each sample was used as a template in a three-step program as follows. Uracil-DNA glycosylase (UDG) activation at 50 °C for 2 min, DNA polymerase activation at 95 °C for 10 min, followed by 40 cycles of DNA denaturation for 15 s at 95 °C, DNA annealing and extension for 1 min at 60 °C, and a last DNA extension step of 15 s at 95 °C. The relative expression level and fold change (FC) of each target gene from resistant field samples relative to the susceptible lab strain were calculated using the 2−ΔΔCT method [30] incorporating the PCR efficiency. Housekeeping genes RPS3 and Actin were used for normalization. 

### 2.5. Data Analysis

#### 2.5.1. Quality Control Filtering and Mapping

De-multiplexed Paired-End (PE) RNA-Seq reads for each sample were evaluated for quality using FastQC v0.11.5 [31]. Subsequently, the PE reads were trimmed and filtered using fastp v0.21.0 [32] to remove adapter and low-quality reads according to the following criteria: minimum base quality score = 20, minimum length required = 25, polyG and poly tail trimming = true. The resulting trimmed and filtered read pairs (R1/R2) were aligned against the reference genome of *Ae. aegypti* (genome assembly version = AaegL5.1, GeneBank assembly identifier = GCA_002204515.1) [33], using ‘*subjunc*’ v2.0.1, part of the subread aligner v2.0.1 [34], with default parameters. The resulting alignment was filtered to remove reads with low mapping quality (q < 10) and sorted successively using Samtools v1.10 [35]. Summary statistics for the QC filtering and sequencing alignments corresponding to malathion, alpha-cypermethrin, and lambda-cyhalothrin are reported in Appendix A. Tags (a read pair or single, unpaired read) mapped to the sense orientation of the annotated *Ae. aegypti* gene set AaegL5.1 were quantified with ‘*FeatureCounts*’, as part of the subread-aligner package v2.0.1 [34], by using the following criteria: (1) count only read pairs that have both ends aligned; (2) count fragment instead of reads; (3) minimum number of overlaps required = 1; (4) feature type = exon; (5) attribute type = gene_id; and (6) strandness = sense. The FeatureCount analysis generated a tag count matrix table which was used as input to edgeR [36] for differential gene expression analysis.

#### 2.5.2. Differential Gene Expression Analysis

For each group, three pairwise comparisons of differential gene expression (DGE) analysis were conducted to identify genes associated with resistance to each insecticide: (1) resistant vs. unexposed control (R–C), (2) resistant vs. ROCK susceptible (R–S), and (3) unexposed control vs. ROCK susceptible (C–S). The R–C comparison was used to detect transcription induced by insecticide exposure, while the R–S and C–S comparisons were used to distinguish genes that are constitutively associated with insecticide resistance as well as those induced by insecticide exposure.

The DGE analysis was conducted using the R package edgeR [36]. To remove the effect of noise and low-expressed genes, for each pairwise comparison, genes with a total tag count less than 50 across all libraries (any pairwise comparison) were filtered out before further analysis as previously suggested [36,37]. The function *calcNormFactors* was used to normalize tag counts among samples, using the TMM (Trimmed Mean M-values) method. This approach normalizes the dataset for sequencing depth and library size, the two most important technical factors that influence DGE analysis. The tag count was not normalized for gene length and GC content, as these values do not vary from sample to sample and would be expected to have little effect on differentially expressed genes (DEGs). The tag-wise and common dispersion of the read count distribution were estimated using the *estimateDisp* function from the edgeR package. The DEGs were selected after multiple testing using the *decideTests* function in the limma package [38]. A critical value absolute log_2_ fold change |log_2_FC| = 1 (|FC| = 2) and a False Discovery Rate (FDR)-adjusted *p* value ≤ 0.01 were used to tag a gene as significantly differentially expressed.

#### 2.5.3. Gene Ontology Annotation and Functional Enrichment Analysis

The *Ae. aegypti* reference genome corresponding to genome assembly version ‘AaegL5.1′ and GeneBank assembly identifier ‘GCA_002204515.1′ (directly downloaded from Vectorbase, Release 51) was used. AaegL5.1 contains 14,718 protein coding genes, 4704 ncRNA genes, and 382 pseudogenes (corresponding to 19,804 genes) [39]. However, functional annotation is provided for only 6397 (36.68%), and gene ontology annotation is provided for only 10,910 (74.13%) genes in VectorBase. To help with the interpretation of our results, predicted genes of AaegL5.1 were functionally annotated using Blast2GO as follows. First, a local BLASTp (v2.9) search of the predicted protein coding sequences was conducted against the Arthropoda (taxid = 6665) category of the nr protein NCBI database with maximum e-value cut-off of 10^−3^. Second, the protein sequences were searched against the InterPro database [40], using InterProScan v5 [41]. The Blastp and InterProScan outputs were simultaneously provided to Blast2GO command line as input, which map the RefSeq and InteProScan identifiers to the GO database as curated and updated in the Blast2GO database (July 2021). This analysis assigned putative descriptions to 13,302 (90.37%) protein coding genes and gene ontology annotation to 11,672 (79.30%), which could be considered a significant improvement. All annotation results (VectorBase and Blast2GO) are provided in Appendix A. GO term enrichment analysis of upregulated and downregulated genes was carried out using Goatools [42], based on the go-basic database (release 1 February 2021). The list of 11,672 blast2GO annotated genes of *Ae. aegypti* with their associated GO terms was used as the background reference set. The *p*-values used to evaluate significantly enriched GO terms were calculated based on Fisher’s exact test and corrected with the Benjamini–Hochberg multiple test correction method [42]. Finally, we used an FDR-adjusted *p*-value of *<* 0.05 to define statistically significant overrepresented (enriched) and underrepresented (purified) GO terms associated with the list of DEGs.

## 3. Results

### 3.1. Bioassay Results

The mosquitoes from both sites showed different patterns of resistance to the three insecticides tested. After 30 min of exposure to alpha-cypermethrin, lambda-cyhalothrin, or malathion, the mortality rate of mosquitoes from Manatí was 46%, 85%, and 56%, respectively, compared to 81%, 48%, and 25%, respectively, of mosquitoes from Isabela (Figure 1, Appendix A).

### 3.2. RNA Sequencing, Quality Control Filtering, and Alignment Rate

The Appendix A shows the statistics of the RNA-Seq sequencing results (before and after quality filtering), the alignment of the filtered reads to the reference genome, and the read quantification. The Illumina sequencing generated approximatively 27.6–69.4 million reads per sample. After adapter trimming and read quality filtering, an average of 41.52 ± 10.43 million reads per sample were retained for subsequent analysis. An average of 24.12 ± 9.43 million quality filtered PE reads per sample were mapped to the reference genome of *Ae. aegypti*, while an average of 7.2 ± 3.03 million tags (read pairs) per sample, representing 50% to 74% of the total alignment, were successfully assigned to the exonic regions of the gene set AaegL5.1 (Appendix A).

### 3.3. Differential Gene Expression Analysis

#### 3.3.1. Differential Gene Expression Associated with Malathion Resistance

For ISA samples, 1132 genes (441 upregulated and 691 downregulated), 563 genes (256 upregulated and 307 downregulated), and 119 genes (33 upregulated and 86 downregulated) were significantly differentially expressed in R–S, C–S, and R–C comparisons, respectively. For MAN samples, a total of 1396 genes (739 upregulated and 657 downregulated), 653 genes (197 upregulated and 456 downregulated), and 215 genes (174 upregulated and 41) were significantly differentially expressed in R–S, C–S, and R–C comparisons, respectively (Table 1, Figure 2, Appendix A). Not surprisingly, the number of DEGs was higher in the R–S comparisons compared to the C–S and R–C comparisons from both sites. However, the difference observed in the malathion resistance between Manatí and Isabela (Figure 1), was not reflected in the numbers of DEGs detected in any of the pairwise comparisons.

The DEGs that overlapped two or more comparisons for each experiment (four intersections in total) were extracted and inspected for important gene expression patterns (Appendix A). For the ISA samples, there were 69 DEGs (24 upregulated and 45 downregulated) that were shared between R–S and R–C groups (Appendix A). The top five shared overexpressed genes between R–S and R–C groups included two odorant binding proteins (OBPs) (the duplicated OBP66), a kDa salivary protein, a zinc finger 512B-like, and an uncharacterized protein (Appendix A). However, for MAN samples, there were 148 differentially expressed genes (126 upregulated and 22 downregulated) that overlapped between R–S and R–C groups (Appendix A); the top 10 shared upregulated genes included seven OBPs (including duplicated OBP66 and OBP67), serine protease snake-like, and two uncharacterized proteins (Appendix A).

Focusing on R–S comparisons, we identified 753 genes (309 upregulated and 444 downregulated) that were differentially expressed in malathion survivors from both ISA and MAN relative to the susceptible ROCK strain (Figure 3A). The top 15 upregulated genes with retrievable annotations included two serine proteases, three uncharacterized proteins, three odorant binding proteins, a 39S ribosomal mitochondrial protein, a digestive organ expansion factor homolog, a kDa secreted-1, a nuclear valosin-containing-like, nucleolar dao-5, peritrophin-1-like, pre-mRNA-processing factor 39, and peritrophin-1-like protein coding genes (Appendix A).

The detoxification genes annotated as cytochrome P450 monooxygenases were predominantly overexpressed, while the cuticular-related protein genes were predominantly downregulated in malathion-resistant samples from both ISA and MAN (Figure 2). Furthermore, 83% (20/24) and 82% (27/33) of differentially expressed CYP450s were upregulated in the R–S comparisons from ISA and MAN, respectively, while 78% (18/23) and 91% (39/33) of cuticular-related DEGs were significantly downregulated in R–S comparisons from ISA and MAN, respectively (Appendix A).

From the list of DEGs shared between the R–S groups from the two sites, we have identified a list of core detoxification genes that are associated with malathion resistance in *Ae. aegypti* in both populations. These core candidate detoxification genes include 22 CYP450s (19 upregulated: CYP6Z7, CYP6D5, CYP6BY1, CYP325Q1, CYP9J2, CYP6N9, CYP6Z6, CYP325V1, CYP6N12, CYP6BB2, CYP6M9, CYP9J23, CYP9J22, CYP9J9, CYP4G36, and four CYP9F2 orthologs; and three downregulated: CYP12F5, CYP325N2, and CYP6N15), one upregulated carboxylesterase, and three glutathione-s-transferases (including one upregulated: GSTD6, and two downregulated: GSTS1 and GSTX1) (Figure 4, Table 2). Although ISA *Ae. aegypti* showed higher resistance to malathion than MAN (Figure 1), we did not detect any detoxification genes that were uniquely overexpressed in malathion-resistant ISA. However, several CYP450s (including CYP4D39, CYP28A5, CYP4J16, and CYP9J23) were overexpressed in malathion-resistant MAN, but not in malathion-resistant ISA. A detailed summary of all the DEGs shared between ISA and MAN is shown in Appendix A. 

#### 3.3.2. Differential Gene Expression Associated with alpha-Cypermethrin Resistance

For ISA, 1171 genes (642 upregulated and 529 downregulated), 478 genes (158 upregulated and 322 downregulated), and 433 genes (87 upregulated and 346 downregulated) were significantly differentially expressed in R–S, C–S, and R–C comparisons, respectively. For MAN, a total of 1175 genes (605 upregulated and 570 downregulated), 811 genes (222 upregulated and 589 downregulated), and 602 genes (154 upregulated and 448 downregulated) were significantly differentially expressed in R–S, C–S, and R–C comparisons, respectively (Table 1, Appendix A). As expected, the numbers of DEGs in the R–S comparisons were higher than the C–S comparisons for both sites. Interestingly, the number of DEGs was higher in the mosquitoes from Manatí, which exhibited higher resistance to alpha-cypermethrin compared to the mosquitoes from Isabela. The list of DEGs that overlapped two or more comparisons for each experiment (four intersections) (Appendix A), were manually extracted and are reported in Appendix A. Unlike the malathion experiment, no DEGs were shared between the R–S and R–C groups. 

Focusing on the R–S comparisons, a total of 801 differentially expressed genes (421 upregulated and 380 downregulated) were shared between the MAN and ISA alpha-cypermethrin survivors, relative to the susceptible strain (Figure 3B). The top 10 overlapping DEGs with retrievable annotation (FC = 15.1-101.6) included a serine protease, a peritrophin-1-like, a kDa secreted salivary protein, an uncharacterized protein LOC5569546 isoform, an uncharacterized protein LOC5569546 isoform X1, NPC intracellular cholesterol transporter, a nuclear valosin-containing-like, a V-type proton ATPase subunit d1, and two cytochrome P450s (CYP67 and CYP9F2) (Appendix A).

Similar to the malathion analysis, the detoxification genes annotated as cytochrome P450 monooxygenases were predominantly overexpressed, while the cuticular protein genes were predominantly downregulated in both alpha-cypermethrin survivors (R–S) and unexposed groups (C–S) relative to the susceptible strain for both study sites (Figure 2. In particular, 76% (22/29) and 74% (20/27) of differentially expressed CYP450s were upregulated in the R–S comparisons for ISA and MAN, respectively, while 97% (34/35) and 97% (33/34) of cuticular related DEGs were significantly downregulated in R–S comparisons for ISA and MAN, respectively (Figure 2 and Appendix A). From the list of DEGs that overlapped the two R–S groups, a list of core detoxification genes associated with alpha-cypermethrin resistance was identified due to their consistent differential expression in both MAN and ISA survivors (Figure 4, Table 2). These candidate alpha-cypermethrin detoxification genes included 21 CYP450s, of which 15 were significantly upregulated (CYP6Z7, CYP9J2, CYP6N12, CYP6BB2, CYP9J22, CYP6N9, CYP6M9, CYP6D5, CYP325V1, CYP6Z6, CYP9J9, CYP4G36, and three CYP9F2 orthologs) and 6 were downregulated (including CYP12F5, CYP6Y3, CYP4AC1, CYP325N, CYP6N15, and CYP4H32) (Figure 4A). Additionally, we identified four GST genes (GSTS1, GSTD1, GSTD11, and GSTE4) that were downregulated in both ISA and MAN alpha-cypermethrin survivors compared to the susceptible strain (Figure 4B).

#### 3.3.3. Differential Gene Expression Associated with Lambda-Cyhalothrin Resistance

For ISA samples, 1261 genes (341 upregulated and 920 downregulated), 480 genes (158 upregulated and 322 downregulated), and 286 genes (30 upregulated and 256 downregulated) were significantly differentially expressed in R–S, C–S, and R–C comparisons, respectively. For MAN samples, a total of 1413 genes (362 upregulated and 1051 downregulated), 811genes (222 upregulated and 589 downregulated), and 254 genes (67 upregulated and 197) were significantly differentially expressed in R–S, C–S, and R–C comparisons, respectively (Table 1 and Appendix A). The number of DEGs in the R–C comparison was higher in the Isabela population, which exhibited higher resistance to lambda-cyhalothrin than the Manatí population.

The DEGs that overlapped two or more comparisons for each experiment were manually extracted and are shown in Appendix A. Although 174 DEGs (8 upregulated and 166 downregulated) and 139 DEGs (12 upregulated and 127 downregulated) were identified in the intersection R–S/R–C, for ISA and MAN, respectively, no upregulated genes were shared between the resistant, susceptible, and control groups from each study site (Appendix A). Among the overexpressed genes in the R–S/R–C intersection for MAN was a glutathione-s-transferase gene (GSTD6) with fold change expression of 10.56 and 2.73 in R–S and R–C, respectively. However, of the eight upregulated genes that overlapped the R–S and C–S groups for ISA, none were previously reported to be associated with insecticide detoxification (Appendix A).

Focusing on only the R–S groups, a total of 943 differently expressed genes (203 upregulated and 743 downregulated) were shared between MAN and ISA lambda-cyhalothrin survivors relative to the susceptible strain (Figure 3C). The top 10 shared upregulated genes (FC = 14.2-1901.7) included a serine protease, a peritrophin-1-like, a digestive organ expansion factor homolog, an uncharacterized protein, nucleolar dao-5, pre-mRNA-processing factor 39, kDa secreted salivary protein, nuclear valosin-containing-like, NPC intracellular cholesterol transporter 2 homolog a, and a CYP450 (CYP6Z7) (Appendix A). Similar to the malathion and alpha-cypermethrin experiments, the detoxification genes annotated as cytochrome P450 monooxygenases were predominantly overexpressed, while the annotated cuticular protein genes were predominantly downregulated in both R–S and C–S groups (Figure 2). In particular, 69% (18/26) and 55% (12/22) of differentially expressed CYP450s were significantly overexpressed in the R–S comparisons from ISA and MAN, respectively, while 97% (34/35) and 97% (33/34) of cuticular-related DEGs were significantly downregulated in R–S comparisons of ISA and MAN lambda-cyhalothrin survivors, respectively (Figure 2 and Appendix A). 

From the DEGs that overlapped the R–S comparisons of both study sites, several core insecticide detoxification genes were identified. These included 17 cytochrome P450s (10 upregulated: CYP6Z7, CYP9J2, CYP28A5, CYP6Z6, two CYP9F2 orthologs, CYP6M9, CYP6BB2, CYP307A1, CYP4C2-like; and 7 downregulated: CYP6P12, CYP6Y3, CYP4H29, CYP6S3, CYP6N15, CYP325N2, CYP4C1-like). Additionally, five GSTs (one upregulated: GSTD6; and four downregulated: GSTX1, GSTE6, GSTD11, and GSTS1) were significantly differentially expressed in mosquitoes that survived lambda-cyhalothrin exposure relative to the susceptible strain for both ISA and MAN (Figure 4, Table 2). 

#### 3.3.4. Genes Associated with Resistance to Multiple Insecticides

A total of 234 genes (43 upregulated and 191 downregulated) were significantly differentially expressed in mosquitoes that survived either malathion, alpha-cypermethrin, or lambda cyhalothrin exposure compared to the susceptible laboratory strain (Figure 5A, Appendix A). Two of the top 10 upregulated DEGs that overlapped all the R–S groups included a putative serine protease (AAEL029072; FC = 1197–6081) and a peritrophin-1-like (AAEL021372; FC = 189–1078), reflecting their potential association with the multiple-insecticide resistance phenotypes (Figure 5B). Eight key genes coding for CYP450 enzymes including CYP6Z7, CYP28A5, CYP9J2, CYP6Z6, CYP6BB2, CYP6M9, and two CYP9F2 orthologs were upregulated across the six R–S comparisons. Two additional CYP450s (CYP325N2, CYP6N15) and one GST (GSTS1) were downregulated in all six resistant groups (Figure 5C). A total of 131, 325, and 240 genes were uniquely differentially expressed in malathion, alpha-cypermethrin, and lambda cyhalothrin survivors, respectively, suggesting their association with a specific insecticide (Figure 5A). Two detoxification DEGs, a CYP450 (CYP6BY1) and a carboxylesterase (AAEL004022), were unique to malathion-resistant samples (Figure 4A). Additionally, one downregulated detoxification gene (GSTD1) was only detected in alpha-cypermethrin-resistant samples. Unique downregulated detoxification genes associated with lambda-cyhalothrin resistance were CYP4H29, CYP307A1, CYP6P12, and GSTE6 (Table 2).

### 3.4. Gene Ontology Enrichment Analysis

Gene ontology enrichment analysis (GOEA) was performed on the list of DEGs associated with all the six R–S comparisons (Appendix A). Not surprisingly, the differences observed in the transcriptomic profiles between the different comparisons were also reflected in the total number of enriched gene ontology (GO) terms associated with DE genes (Table 1 and Appendix A). GO analysis of the upregulated genes for the R–S comparisons, identified the enrichment of some relevant molecular functions that are strongly associated with multiple-insecticide resistance phenotypes, including “monooxygenase activity” (GO:0004497), “oxidoreductase activity” (GO:0016705), “heme binding” (GO:0020037), and “tetrapyrrole binding”(GO:0046906) (Figure 6). Additionally, the molecular functions “catalytic activity” (GO:0003824) and “iron ion binding” (GO:0005506) were enriched in four out of six upregulated gene lists analyzed, reflecting their association with multiple-insecticide resistance. Interestingly, the enrichment of some relevant GO terms was found to be specific to malathion-resistant mosquitoes, including several GO terms associated with postsynaptic neurotransmitter activities (GO:1904315, GO:0099529, GO:0098960, GO:0030594), transmembrane transporter activities (GO:0022803, GO:002285, GO:0022857), ion channel activities (GO:0005216, GO:0005261, GO:0015267, GO:0022848, GO:0099094, GO:0005231, GO:0022848), and protein receptor activities (GO:0004930, GO:0005102) (Figure 6). For the list of downregulated genes, several GO terms were overrepresented in four out of the six R–S comparisons. These included “catalytic activity” (GO:0003824), “carboxypeptidase activity” (GO:0004180), “metallocarboxypeptidase activity” (GO:0004181), “structural constituent of chitin-based cuticle” (GO:0005214), “chitin binding” (GO:0008061), and “exopeptidase activity” (GO:0008238) (Appendix A). GO terms associated with chitin binding activities were overrepresented in the list of downregulated genes of all the six R–S DEG comparisons, which was also clearly reflected in the functional annotation of the DEGs displayed in Figure 2, where genes associated with chitin and chitinase activities were mostly downregulated (Appendix A).

### 3.5. Validation of Relative Expression Levels Estimated Using RNA-Seq with Quantitative RT-PCR

Five transcripts differentially expressed in resistant samples (I-ACYP, I-LCT, and M-LCT), including GSTD6, GSTE4, CYPBB2, CYP6M9, and CYP6N9, were used to validate the RNA-Seq results with quantitative RT-PCR. The threshold cycle (ct) value was not detected for 2 of the 15 qPCR assays, likely associated to the low expression level of these transcripts in the mosquitoes. This is evident in the low read count from the RNA-Seq. The qRT-PCR results broadly supported the directionality of the changes in expression levels (75% of the essays with ct values), although for several genes, the magnitude of the expression difference was smaller than estimated using RNA-Seq (Appendix A).

## 4. Discussion

Insecticide resistance in *Ae. aegypti* continues to expand globally due the extensive use of insecticides for its control. In this study, we used whole transcriptomic approach to investigate the molecular basis of resistance in *Ae. aegypti* populations from two locations in Puerto Rico, Isabela and Manatí, that were resistant to an organophosphate (malathion) and two pyrethroids (lambda-cyhalothrin and alpha-cypermethrin).

Differential gene expression analysis of genes associated with resistance to all the three insecticides from the two locations showed similarities in overexpression of the detoxification genes belonging mainly to the cytochrome P450 gene family and downregulation in cuticular protein genes. The insect cuticle is comprised mainly of chitin and cuticular proteins, and modifications to the cuticle can lead to thickening of the cuticle hence slowing the penetration of insecticides [43]. Cuticular thickening has been associated with resistance to pyrethroids in *Anopheles funestus* [44] and with reduced penetration of deltamethrin in highly resistant strain of *Anopheles gambiae* [45]. The downregulation of cuticular proteins in this study suggests that resistance in these populations is predominantly driven by cytochrome P450-mediated detoxification of all three insecticides. A study on the transcriptomic profile of a resistant strain of *Ae. aegypti* from Vietnam similarly reported downregulation of cuticular proteins in resistant versus susceptible strains [18]. 

In all the resistant groups of all three insecticides, a serine protease (AAEL029072) was significantly upregulated with a high fold change (Figure 5B). Serine protease has been reported to be highly upregulated in *An. gambiae* resistant to DDT [46] as well as in *Culex pipiens pallens* resistant to deltamethrin [47], where it was suggested to play a crucial role in the innate immunity in this species and other insects [48,49,50]. This finding suggests that the increase in protease activity may be important in modulating *Ae. aegypti* resistance to multiple insecticides; however, this remains speculative as this has not yet been functionally proven.

Several CYP450s were also commonly differentially expressed in the three resistant groups. These included CYP6Z7, CYP28A5, CYP9J2, CYP6Z6, CYP6BB2, CYP6M9, and two CYP9F2 orthologs, suggesting that these CYPs may confer cross-resistance to pyrethroids and organophosphates. The majority of these CYPs have been associated with resistance to insecticides in previous studies. CYP6BB2 has consistently been reported to be overexpressed in *Ae. aegypti* resistant to pyrethroids in Asia, Europe, and the Americas [51,52,53,54,55]. The overexpression of CYP6BB2 has also shown strong metabolic activity for permethrin in in vitro metabolic studies [51], highlighting its importance as a candidate metabolic resistance marker for pyrethroids in *Ae. aegypti* populations. In the same study, CYP6Z7 and CYP6Z6 were also overexpressed in permethrin-resistant *Ae. aegypti*. In another study, CYP6Z7 and CYP6BB2 were among four of the only CYPs upregulated in three tested strains of *Ae. Aegypti* resistant to deltamethrin while CYP9J2 also appeared in one of the resistant strains in the same study [56]. Multiple studies have also reported the upregulation of CYP6Z6 in pyrethroid-resistant *Ae. aegypti* populations around the world [52,53,54,57,58]. 

Despite some shared gene expression patterns across the three insecticides, some of the genes were only found to be differentially regulated in survivors of specific insecticides and not others. In the malathion-resistant samples from both locations, CYP6BY1 was found to be upregulated but was not upregulated in the lambda-cyhalothrin or alpha-cypermethrin-resistant samples. CYP6BY1 has previously been reported to be among only two overexpressed P450s in Colombian *Ae. aegypti* larvae resistant to temephos, which is also an organophosphate insecticide [59]. In samples resistant to alpha-cypermethrin, the glutathione S-transferase gene GSTD1 was found to be uniquely overexpressed in this group compared to survivors of the other two insecticides. GSTD1 has been implicated in resistance to DDT in both *An. gambiae* [60] and *Drosophila melanogaster* [61]. Two genes, CYP4H29 and GSTE6, were uniquely overexpressed in samples resistant to lambda-cyhalothrin. In a targeted RNA-Seq study, GSTE6 was the only GST reported to be associated with deltamethrin resistance across eight *Ae. aegypti* strains from distinct geographical origins [62]. CYP4H29 has previously been associated with resistance to pyrethroids in Puerto Rican *Ae. aegypti* [54]. The same study reported the overexpression of four CYP450s belonging to family four. Studies on the role of cytochrome P450 4 (CYP4) family in resistance are scarce; however, Reid et al. showed increased survival of *D. melanogaster* expressing CYP4H29 when exposed to permethrin [54]. 

Another difference that stands out is the enrichment of GO terms belonging to the postsynaptic neurotransmitter receptor activity mainly in the alpha-cypermethrin-resistant samples from both Isabela and Manatí. This supports the evidence that type II pyrethroids such as alpha-cypermethrin have been shown to mainly inhibit GABA_A_ receptor function, hence, leading to higher excitation in the insect’s central nervous system [63]. However, the same GO terms did not appear enriched in the lambda-cyhalothrin resistant group, which is also a type II pyrethroid. According to the WHO report 1998 [64], alpha-cypermethrin was more toxic than other pyrethroids and was significantly more effective at killing anopheline mosquitoes than permethrin or lambda-cyhalothrin, and this may explain the high activity in the enriched GO terms. In addition, the bottle bioassay results showed a higher frequency of resistance to alpha-cypermethrin in the Manatí population than in the Isabela population, potentially also explaining the higher enrichment score in this group.

## 5. Conclusions

This study investigated the gene expression profiles associated with pyrethroid and organophosphate resistance in *Ae. aegypti* populations from two locations in Puerto Rico. We identified several detoxification genes that exhibited similar differential overexpression patterns across all the insecticide-resistant phenotypes from both populations, suggesting that these genes could be used as expression-based markers for multiple-insecticide resistance in *Ae. aegypti*. We also identified some detoxification genes that were uniquely overexpressed in mosquitoes that survived exposure to malathion, alpha-cypermethrin, or lambda-cyhalothrin, indicating that their overexpression is more closely associated with resistance to specific insecticides. Additionally, we report the association of significant overexpression of serine protease genes and downregulation of cuticular-related protein genes (chinase and chitin) with organophosphate and pyrethroid resistance in both *Ae. aegypti* populations. The genes identified in this study, particularly in the cytochrome P450 family, should be functionally validated to confirm their importance as molecular markers for resistance detection.

## Figures and Tables

**Figure 1 genes-14-01626-f001:**
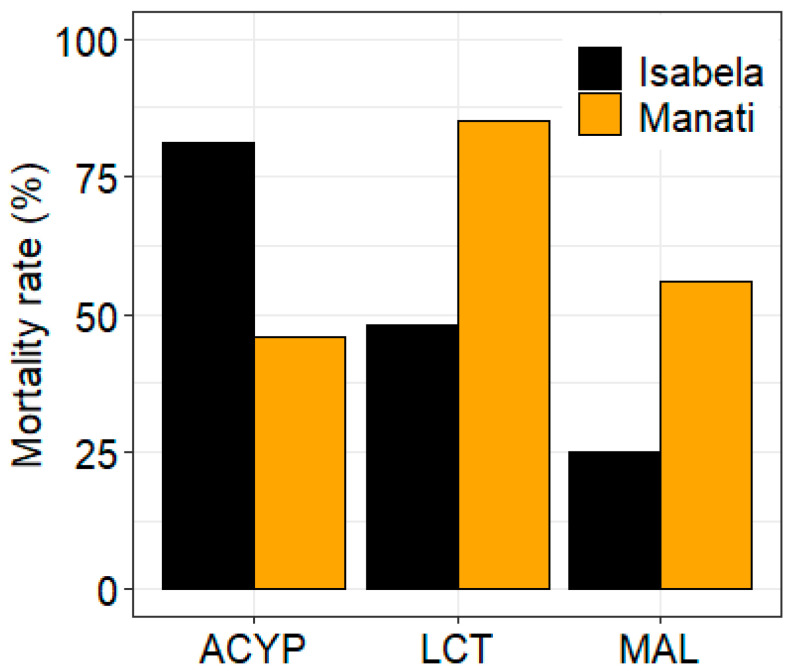
Bioassay results for alpha-cypermethrin (ACYP), lambda-cyhalothrin (LCT), and malathion (MAL) insecticides for *Ae. aegypti* from Isabela and Manatí in Puerto Rico. Bars show the percent mortality after 30 min of insecticide exposure.

**Figure 2 genes-14-01626-f002:**
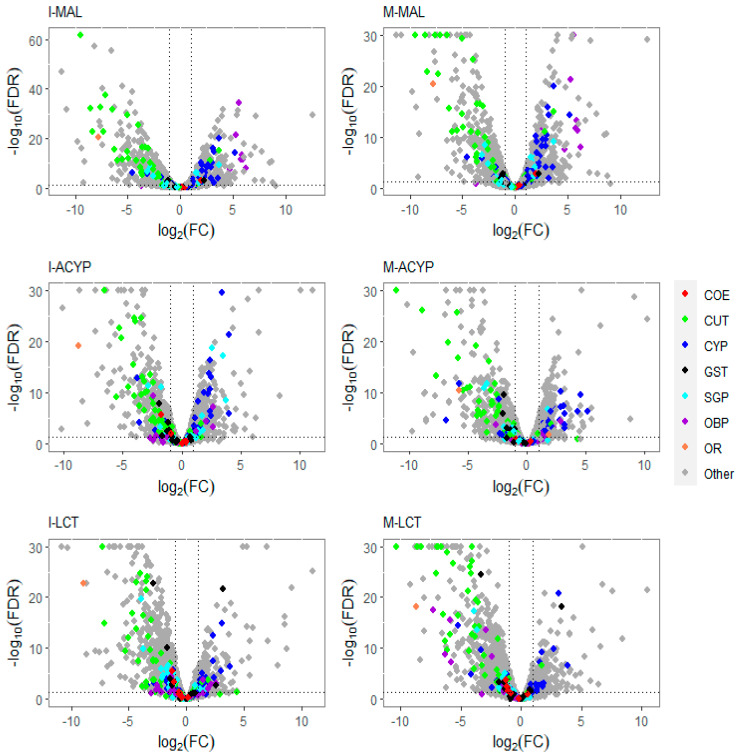
Volcano plots of gene expression profiles of mosquitoes that survived insecticide exposure compared to the susceptible laboratory strain (ROCK). The level of gene expression is plotted in the *x*-axis as the log_2_ fold change, and the significance the statistical test is plotted in the *y*-axis as -log_10_ of the corrected *p*-value (−log_10_FDR values greater than 50 are displayed as 30). The panels include gene expression profiles compared to ROCK for Isabela malathion (I-MAL), Manatí malathion (M-MAL), Isabela alpha-cypermethrin (I-ACYP), Manatí alpha-cypermethrin (M-ACYP), Isabela lambda-cyhalothrin (I-LCT), and Manatí lambda-cyhalothrin (M-LCT). Detoxification gene families are denoted in red (COE: carboxylesterases), blue (CYP: cytochrome P450s), and black (GST: glutathione-S-transferases). Cuticular proteins (CP) are shown in green, salivary gland proteins (SGP) in cyan, odorant binding proteins (OBP) in violet, and odorant receptor genes in orange. In each plot, genes overexpressed in the resistant groups have a log_2_FC > 0. The vertical dotted lines indicate two-fold change expression differences, and the horizontal lines indicate an FDR of 0.01.

**Figure 3 genes-14-01626-f003:**
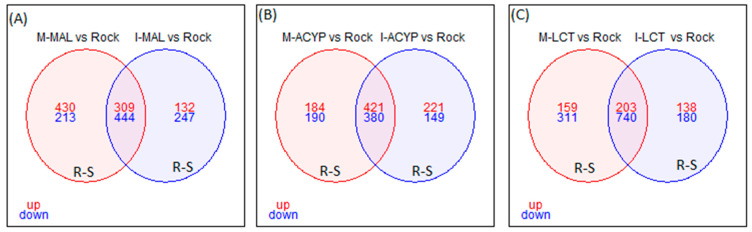
Venn diagram of differentially expressed genes (log2FC > 1 and FDR < 0.01) in insecticide-resistant mosquitoes compared to the susceptible laboratory colony. (**A**) Malathion-resistant samples from Manatí (M-MAL) and Isabela (I-MAL); (**B**) alpha-cypermethrin-resistant samples from Manatí (M-ACYP) and Isabela (M-ACYP); and (**C**) Lambda-cyhalothrin resistant samples from Manatí (M-LCT) and Isabela (I-LCT). The overlapping areas represent genes that that are differentially expressed across both sites. Upregulated (up) genes are in red and downregulated (down) genes are in blue.

**Figure 4 genes-14-01626-f004:**
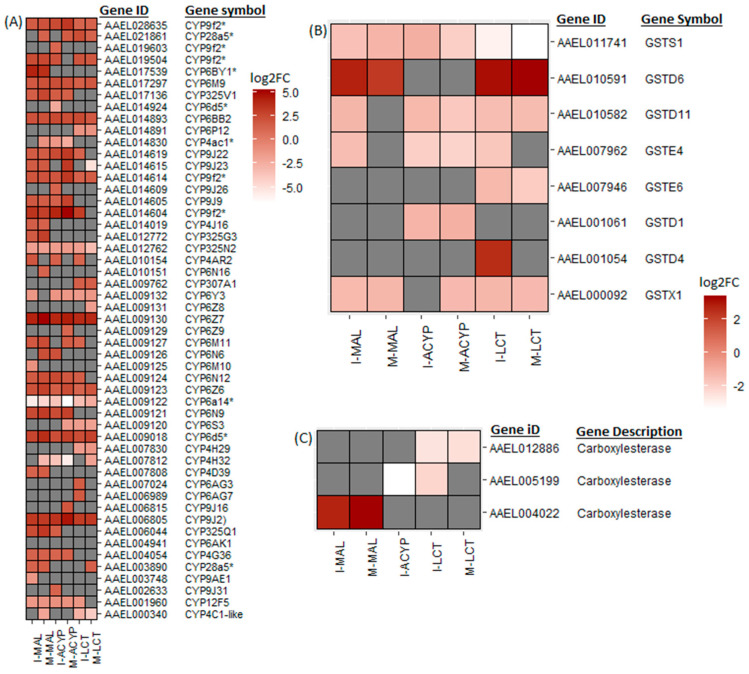
Heatmaps summarizing the log2 fold change in expression of detoxifications genes including cytochrome P450s (**A**), glutathione-S transferases (**B**), and carboxylesterases (**C**). The log2FC expression is shown on a white–red scale (red = overexpressed, white = under-expressed, and gray = not detected or not differentially expressed) for the comparisons of insecticide-resistant samples vs. susceptible lab strain (R–S). The asterisks (*) are used to identify gene symbols that were assigned through Blast2GO annotation.

**Figure 5 genes-14-01626-f005:**
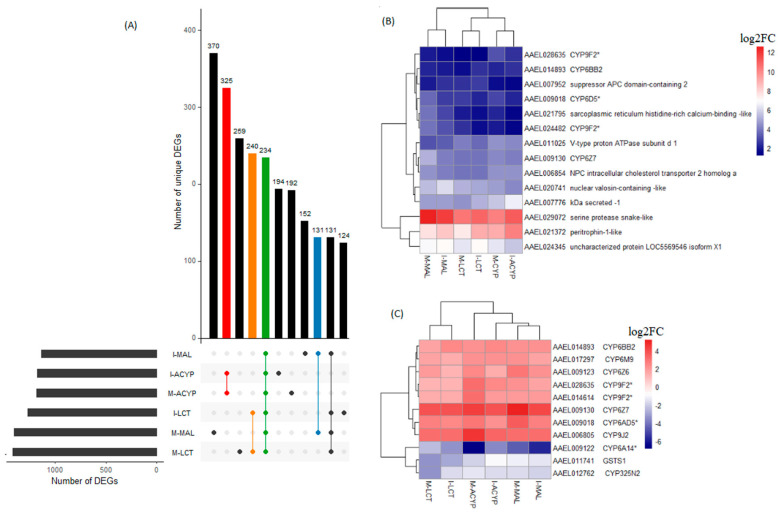
Identification of DEGs associated with multiple-insecticide resistance. (**A**) The upset plot represents the intersection of differentially expressed genes from Isabela (I) and Manatí (M) between the malathion (MAL), alpha-cypermethrin (ACYP), and lambda-cyhalothrin (LCT) resistant samples when compared to the insecticide susceptible strain (R–S). The left horizontal bar plot reports the total number of DEGs in each comparison (set size), and the circles represent the set of comparisons associated to the intersection, while the vertical bar plot reports the number of unique and overlapping DEGs (intersection size) between the different combinations of R–S comparisons. The bar plots represent core DEGs (green) and DEGs specific to MAL (blue), ACYP (red), and LCT (orange). (**B**) Heatmap showing the log2 fold change (log_2_FC) expression of the top 10 DEGs in any R–S comparison (core top 10). (**C**) Heatmap showing the log2FC expression of the detoxification genes shared across all R–S comparisons. The heatmaps are in a blue–red color gradient (red = overexpressed and blue = under-expressed). The asterisks (*) are used to identify gene symbols that were assigned through Blast2GO annotation.

**Figure 6 genes-14-01626-f006:**
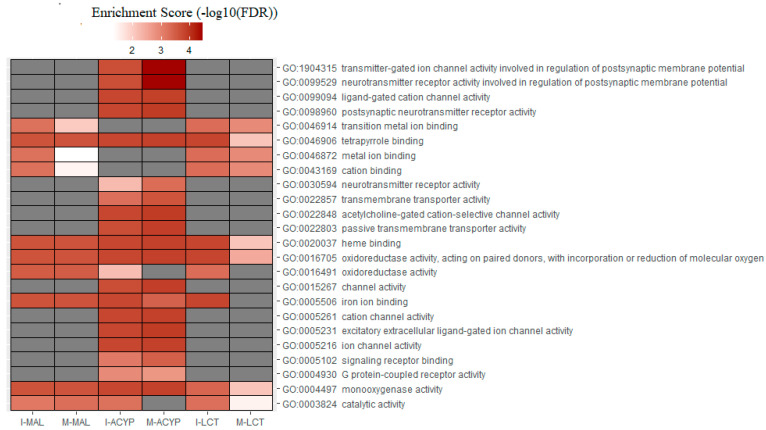
Heatmap of the gene ontology (GO) enrichment analysis of the significantly upregulated genes in the malathion (I-MAL and M-MAL), alpha-cypermethrin (I-ACYP and-M-ACYP), and lambda-cyhalothrin (I-LCT and M-LCT) resistant samples compared to the susceptible strain of *Ae. aegypti*. The white–red gradient indicates the enrichment score of each GO term, while the gray color indicates that the GO term was not significantly enriched. Only the GO terms that overlapped for both MAN and ISA for specific insecticides are shown. The complete GOEA report is shown in Appendix A.

**Table 1 genes-14-01626-t001:** Summary of the total number of differentially expressed genes detected for malathion, alpha-cypermethrin, and lambda-cyhalothrin comparisons: R–S (resistant vs. susceptible), C–S (unexposed control vs. susceptible), or R–C (resistant vs. unexposed control).

			# Of Genes Tested	DE Genes(|log_2_FC| > 1 and FDR < 0.05)		DE Genes(|log_2_FC| > 1 and FDR < 0.01)	
Insecticide	Sites	Comparisons		Up	Down	Up	Down
Malathion	Isabela	I-MAL vs. Rock (R–S)	10,153	699	939	441	691
		I-MAL vs. I-U1 (R–C)	10,161	120	186	33	86
		I-U1 vs. Rock (C–S)	10,088	337	403	256	307
	Manatí	M-MAL vs. Rock (R–S)	10,573	961	792	739	657
		M-MAL vs. M-U (R–C)	10,885	322	98	174	41
		M-U vs. Rock (C–S)	10,039	333	634	197	456
alpha-cypermethrin	Isabela	I-ACYP vs. Rock (R–S)	9825	711	625	642	529
		I-U vs. Rock (R–C)	9711	230	426	158	322
		I-ACYP vs. I-U (R–C)	9308	127	426	87	346
	Manatí	M-ACYP vs. Rock (R–S)	9599	686	692	605	570
		M-ACYP vs. M-U (R–C)	7835	237	561	154	448
		M-U vs. Rock (C–S)	9561	301	758	222	589
Lambda-cyhalothrin	Isabela	I-LCT vs. R vs. Rock (R–S)	9916	429	1008	341	920
		I-U vs. Rock (C–S)	9711	230	426	158	322
		I-LCT vs. I-U (R–C)	9550	51	366	30	256
	Manatí	M-LCT vs. Rock (R–S)	9710	461	1163	362	1051
		M-U2 vs. Rock (C–S)	9561	301	758	222	589
		M-LCT vs. M-U (R–C)	8618	126	271	67	197

**Table 2 genes-14-01626-t002:** Fold change expression of significantly differentially expressed genes of interest (|logFC| > = 1, FDR < = 0.01) in the comparison of resistant vs. susceptible (R–S) and unexposed control vs. susceptible (C–S) groups for malathion (MAL), alpha-cypermethrin (ACYP), and lambda-cyhalothrin (LCT) for the two locations (I: Isabela, M: Manatí). U_1_ and U_2_ denote the unexposed (control) groups used for the MAL and pyrethroid (ACYP and LCT) comparisons, respectively. The asterisks (*) are used to identify gene symbols that were assigned through Blast2GO annotation.

Gene ID	Description	I-MAL vs. Rock (R–S)	I-U1 vs. Rock (C–S)	M-MAL vs. Rock (R–S)	M-U1 vs. Rock (C–S)	I-ACYP vs. Rock (R–S)	I-LCT-R vs. Rock (R–S)	I-U2 vs. Rock (C–S)	M-ACYP vs. Rock (R–S)	M-LCT vs. Rock (R–S)	M-U2 vs. Rock (C–S)
** *Cytochrome P450s monooxygenases* **											
AAEL009130	CYP6Z7	17.19	17.45	36.13	14.77	15.5	13.63	14.92	19.44	14.69	
AAEL006805	CYP9J2	9.25	8.16	8.8	11.56	10.58	8.29	7.15	23.33	8.87	8.96
AAEL009121	CYP6N9	5.57	3.17	8.42	4.5	5.16			7.28		
AAEL006044	CYP325Q1	5.18	3.23	9.48	3.23	2.77		3.55			
AAEL014893	CYP6BB2	4.27	7.01	4.37	6.73	5.48	5.34	6.01	4.71	3.37	3.75
AAEL009123	CYP6Z6	4.22		7.4		2.81	2.31		5.14	3.69	
AAEL010154	CYP4AR2	3.62	2.84			2.54	2.24	2.35			
AAEL014605	CYP9J9	3.43	2.47	2.91	3.32	2.57		2.2	6.79		2.62
AAEL017297	CYP6M9	3.35	6.91	4.36	4.67	4.84	2.68	5.39	4.15	3.41	3.38
AAEL009127	CYP6M11	3.34		5.13	3.69		2.23		2.96		
AAEL012772	CYP325G3	3.19		7.72							
AAEL009124	CYP6N12	3.18	5.06	4.71	4.18	5.56	2.86	3.91	3.45		
AAEL014619	CYP9J22	3.13	2.44	3.09		5.27	2.39	2.27	8.75		
AAEL014615	CYP9J23	3.08		3.31					9.06	0.03	0.3
AAEL014019	CYP4J16	2.86		2.54							
AAEL017136	CYP325V1	2.79	2.95	6.14		2.94			2.68		
AAEL007808	CYP4D39	2.38		2.91							
AAEL004054	CYP4G36	2.37	2.13	2.54		2.18			2.57		
AAEL009125	CYP6M10	0.46									0.4
AAEL009132	CYP6Y3	0.39			0.3	0.41	0.34	0.45	0.38	0.38	0.34
AAEL003748	CYP9AE1	0.38									
AAEL012762	CYP325N2	0.28		0.29	0.22	0.31	0.3		0.34	0.1	0.22
AAEL009762	CYP307A1						2.73			2.71	2.38
AAEL014609	CYP9J26		2.73		3.04	2.23		2.77			2.02
AAEL004941	CYP6AK1				0.38						0.47
AAEL007010	CYP6AG4										0.34
AAEL009120	CYP6S3				0.28		0.31		0.33	0.27	0.22
AAEL007812	CYP4H32		0.12	0.11	0.05	0.08		0.1	0.02	0.26	0.12
AAEL006989	CYP6AG7				3.24		3.24				
AAEL009126	CYP6N6		3.49	4.87		4.3		3.65			
AAEL010151	CYP6N16			2.31							
AAEL014891	CYP6P12						0.44			0.45	
AAEL007830	CYP4H29						0.32			0.38	
AAEL009131	CYP6Z8									0.3	
AAEL006815	CYP9J16								3.58		
AAEL009129	CYP6Z9								2.13		
AAEL002633	CYP9J31					2.27					
AAEL007024	CYP6AG3						2.88				
AAEL026582	CYP6AA5		2.17		2.08						
AAEL001960	CYP12F5	0.37		0.42		0.49	0.4	0.38	0.47		
AAEL014604	CYP9f2*	10.73	12.18	7.65	14.15	16.08	8.03	12.05	36.68		
AAEL009018	CYP6d5*	6.24	3.07	12.91	4.33	4.81	4.94	3.56	7.26	6.66	5.05
AAEL019504	CYP9f2*	5.12	4.14	5.81	6.98	4.16	3.89	4.04		3.43	
AAEL014614	CYP9f2*	3.81	4	3.96	5.69	3.72	2.54	3.05	9.18	2.92	4.28
AAEL028635	CYP9f2*	3.16	3.98	4.18	5.02	5.34	2.3	3.15	9.43	2.45	4.78
AAEL003890	CYP28a5*	2.39		2.82						2.67	
AAEL009122	CYP6a14*	0.02	0.07	0.04	0.09	0.08	0.1	0.01	0.01	0.19	0.11
AAEL021861	CYP28a5*		2.89	2.4			5.03	3.85	3.02	2.37	2.32
AAEL014830	CYP4ac1*			0.47	0.29	0.37			0.19		0.33
AAEL000340	CYP4C1-like			0.24			0.15			0.06	0.19
AAEL019603	CYP9f2*					2.05					
AAEL014924	CYP6d5*					0.21					
AAEL017539	CYP6BY1*	17.22		12.48							
** *Glutathione S-transferases* **											
AAEL010591	GSTD6	7.03	5.58	4.81	3.98		8.87	5.23		10.56	
AAEL010582	GSTD11	0.43			0.33	0.41	0.37	0.41	0.3	0.38	
AAEL000092	GSTX1	0.4		0.42			0.42		0.39	0.39	
AAEL007962	GSTE4	0.37	0.27		0.23	0.26	0.31	0.24	0.24		0.21
AAEL011741	GSTS1	0.35		0.44	0.37	0.49	0.13		0.26	0.1	0.29
AAEL007946	GSTE6				0.24		0.41			0.28	0.18
AAEL001061	GSTD1					0.44			0.48		
AAEL001054	GSTD4						5.97	8.34			
** *Carboxylesterases* **											
AAEL004022	CES5A	3.19		4.07							
AAEL005199	CES6-like		0.36		0.33	0.3	0.48				0.24
AAEL005200	CES6-like									0.48	
AAEL012886	CES-6						0.41			0.43	

## Data Availability

Sequence data generated by this study is available at Sequence Read Archive (SRA) under the Bio Project accession number PRJNA801226.

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
