# Peer review of "Comparative Transcriptomic Analysis of Insecticide-Resistant Aedes aegypti from Puerto Rico Reveals Insecticide-Specific Patterns of Gene Expression"

_genes, 2023, doi:10.3390/genes14081626_

Round 1
Reviewer 1 Report
Dear authors, thank you for your article, it is interesting, but I have a few suggestions to improve its quality and content.
Your introduction should include one to two sentences to justify your choice of your insecticides (malathion, alpha-cypermethrin, and lambda-cyhalothrin) and your study site (Puerto Rico).
Methods :
L105 : It looks like you assessed the mortality rate after 30 minutes. The World Health Organization recommends assessing the mortality rate after 24 hours. Please could you clarify why did you use 30 minutes?
L141. Your sentence is incomplete.
L152 : A few citations are not included in your references. Example : Schmittgen et al 2008, Smith et al 2018 and Mitra et al 2021.
L162 : You should use the most recent version of Aedes aegypti genome for optimizing your gene identification and probably the mapping rate
Results :
There are several error messages that need to be fixed. For instance, L217-218 and L233.
You should include the ID of your differentially expressed genes (DEGs) (AAEL…..).
You should include an interpretation of your bioassay results and your DEGs.
L243 : Your mapping rate is low, did you use another mapper like STAR to verify your mapping rate.
L620-623: The name of funder agency is absent.
You should double check your Informed Consent Statement.
Author Response
Dear reviewer,
Thank you for giving us the opportunity to submit a revised draft of the manuscript “Comparative Transcriptomic Analysis of Insecticide Resistant Aedes aegypti from Puerto Rico Reveals Insecticide-Specific Patterns of Gene Expression”. We appreciate the time and effort you dedicated to providing feedback on our manuscript and are grateful for the insightful comments that help improving the manuscript. Please see the attachment bellow, in blue for a point-by-point response to your comments, suggestions and concerns.
Cordially,
~DD

Reviewer 2 Report
Dear Authors,
The article is very interesting and furnishes useful information that certainly deserves to be highlighted and shared. Under a technical point of view the work done seems to me very good but I am not fully convinced by the experimental design of the susceptibility tests and by the choice of the groups to be studied in the gene expression studies. It is not clear whether the susceptible strain had been tested for its resistance to the insecticides. Have all the individuals from the susceptible strain been confirmed as susceptible? Otherwise, would have been opportune to ascertain and show it to use this strain as control. In the absence of insecticide-specific tests, the individuals of the ROCK strains are only assumed to be susceptible. Also, certain detoxification pathways are only activated by the presence of an active substance. Therefore, in my opinion two different negative control should have been necessary: i) untreated individuals; ii) treated individuals that are not resistant (i.e., the individuals killed by the insecticide).
In the absence of this double control, the conclusions seem to me not sufficiently supported by the data because we do not know whether the detoxification pathways can be also present in the susceptible strain (because it has not been treated) and we do not know whether the death of the treated individuals is due to the fact that this pathways are not active (because these individuals have not been tested in the gene expression studies).
Consequently, in my opinion, the article need a clarification about the experimental design and taking into account these comments in discussion and conclusions.
Other minor points:
Paragraph 3.1. and Fig. 1. It would be opportune to also add a control, which are the result for the susceptible strain? This comparison could add information on the variability of this adaptative trait.
Figure 3. I suggest using the same id for the populations, ISA per Isabela and MAN per Manatì or to better explain in the caption.
Kind regards
Author Response

(The authors gave the same response as above.)

Round 2
Reviewer 1 Report
Not more comments
Reviewer 2 Report
Dear Authors,
thank you for your reply to my comments. I understand your point of view. Indeed the degradation of the RNA could not be controlled under the experimental conditions and it could differ individual by individual. However, ROCK individuals exposed to the insecticides an killed under controlled conditions could have been used for the purpose. Anyway, I suppose it is too late to consider this option and, even if I would have considered it as an important addition I think that the article is sufficiently rich in information to be suitable for publication.
Kind regards